# Learning the Interaction Prior for Protein-Protein Interaction Prediction: A Model-Agnostic Approach

Ziqi Gao [* 1]  Chenyi Zi [* 2]  Zijing Liu [3]  Ziqiao Meng [4]  Yu Li [3]  Jia Li [† 2]

## Abstract

Protein-protein interactions (PPIs) are fundamental to cellular function and disease mechanisms. Current learning-based PPI predictors focus on learning powerful protein representations but neglect designing specialized classification heads. They mainly rely on generic aggregating methods like concatenation or dot products, which lack biological insight. Motivated by the biological "L3 rule", where multiple length-3 paths between a pair of proteins indicate their interaction likelihood, our study addresses this gap by designing a biologically informed PPI classifier. In this paper, we provide empirical evidence that popular PPI datasets strongly support the L3 rule. We propose an L3-path-regularized graph prompt learning method called **L3-PPI**, which can generate a prompt graph with virtual L3 paths based on protein representations and controls the number of paths. L3-PPI reformulates the classification of protein embedding pairs into a graph-level classification task over the generated prompt graph. This lightweight module seamlessly integrates with PPI predictors as a plug-and-play component, injecting the interaction prior of complementarity to enhance performance. Extensive experiments show that L3-PPI achieves superior performance enhancements over advanced competitors.

## 1. Introduction

Protein-Protein Interactions (PPIs) are fundamental processes where proteins bind (Gromiha et al., 2017; Nooren & Thornton, 2003) and perform essential biological functions (Franceschini et al., 2012). Understanding and pre-

---
[*]Equal contribution . [†]Corresponding author. [1] Tsinghua University [2]The Hong Kong University of Science and Technology (Guangzhou) [3] IDEA Research [4] National University of Singapore. Correspondence to: Jia Li <jialee@hkust-gz.edu.cn>.

*Proceedings of the 43rd International Conference on Machine Learning*, Seoul, South Korea. PMLR 306, 2026. Copyright 2026 by the author(s).

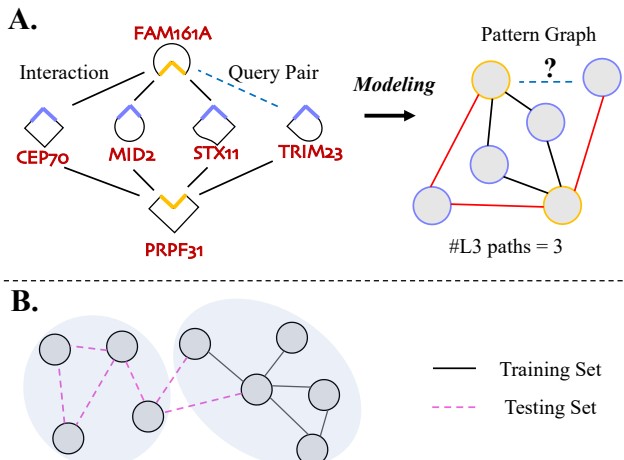

*Figure 1.* (A). **The L3 rule**. This illustrates a typical pattern in PPI networks: two proteins, FAM161A and PRPF31, with similar interfaces, share many common neighbors. In a pattern graph, multiple L3 paths (e.g., the red path) appear between the query proteins, suggesting a high probability of PPI. (B). **The limitation of using #L3 paths as a handcrafted feature**. If the training and testing clusters have low connectivity, almost all testing samples lack L3 paths.

dicting PPIs is critical for advancing novel therapeutic design (Gane & Dean, 2000; Wirthmueller et al., 2013; Wu et al., 2011) and genetic research (Guan et al., 2014; Oti et al., 2006), as they underpin vital cellular activities. Traditional experimental methods, including X-ray crystallography (Kortemme et al., 2004; Engh & Huber, 1991), cryo-EM (Schmidt & Urlaub, 2017; Su et al., 2022) and yeast two-hybrid screens (Fields & Song, 1989), provide valuable insights but are constrained by high costs, time intensity, and limited scalability. Recently, deep learning (DL) has revolutionized the PPI prediction. By extracting interaction rules from sequences (Chen et al., 2019), structures (Bryant et al., 2022), and PPI network topology (Lv et al., 2021), DL methods have made significant progress in computational efficiency and generalization for PPI prediction.

DL-based predictors excel at extracting expressive protein representations using PPI-specialized architectures, including convolutional neural networks (CNNs) (Zhang et al., 2019), recurrent neural networks (RNNs) (Chen et al., 2019),

graph neural networks (GNNs) (Lv et al., 2021; Zhao et al., 2023; Gao et al., 2023b; Tang et al., 2022; Niu et al., 2025), and protein language models (PLMs) (Jumper et al., 2021; Hayes et al., 2025). Nevertheless, few studies address the design of classification heads for PPI prediction, specifically on how to effectively leverage the extracted powerful protein representations. Most existing approaches adopt standard aggregators from graph link prediction work (Zhang & Chen, 2018; Zhu et al., 2021), including concatenation, Hadamard product, or summation of a pair of vector representations. However, these fail to explicitly capture *complementarity priors*, including the geometric complementarity (Gabb et al., 1997; Grünberg et al., 2004) and chemical compatibility (McCoy et al., 1997; Nooren & Thornton, 2003) critical at interaction interfaces. Since complementarity is inherently abstract and difficult to quantify within DL frameworks, a key challenge emerges:

*(Q) How can we incorporate complementarity priors in PPI prediction models?*

To fill this gap, we reference the ***L3 rule*** (Kovács et al., 2019; Yuen & Jansson, 2023; Chen et al., 2020), which quantifies protein complementarity using path features in the PPI network. This rule confirms that the *pattern graph* illustrated in Figure 1.A is widely observed from both PPI network and evolutionary perspectives. In this *pattern graph*, two proteins with similar interfaces share many of their neighbors. Put differently, a greater number of length-3 paths between a protein pair suggests a higher likelihood of interaction. However, directly incorporating the number of L3 paths (#L3 paths) as a handcrafted feature in DL models is challenging. This limitation arises because under commonly used rigorous data splits (Lv et al., 2021; Wu et al., 2024; Gao et al., 2023a), testing protein pairs may be disconnected from the main PPI network (see Figure 1.B), preventing accurate quantification of #L3 paths. To overcome this issue, our core idea is to generate a *pattern graph* between them, where #L3 paths correspond to the interacting strength. The graph-level classification of the generated *pattern graph* is then used for the final PPI prediction.

In this work, we first offer more extensive empirical evidence for the L3 rule compared to Kovács et al. (2019). We test two commonly used PPI prediction datasets, assessing both Pearson correlation and mutual information between #L$i$ paths ($i = 2, 3, ..., 7$) and interaction labels. We endow prevailing DL-based PPI prediction models by introducing complementarity-induced classification heads. For this purpose, we propose **L3-PPI**, a graph prompt learning method that strategically incorporates the L3 rule. L3-PPI initially generates and evaluates potential L3 paths through a specialized gating network. The network then regularizes the number of activated paths based on the PPI label of a protein pair, favoring more activated paths for positive

PPIs. Finally, the activated paths collectively form a *pattern graph*, and we reformulate the PPI classification task for a pair of proteins as a graph-level classification task of their *pattern graph*. Extensive evaluations show that L3-PPI consistently enhances the performance of existing PPI prediction methods, achieving state-of-the-art results across multiple benchmarks.

## 2. Related Work

**Protein-Protein Interaction (PPI).** PPI prediction aims to identify both interacting protein pairs and their specific interaction types. Early machine learning (ML) methods primarily rely on amino acid sequences information. These approaches, employing ML algorithms like Support Vector Machines and Rotation Forest, focus on designing effective sequence encoding techniques (Wong et al., 2015; Shen et al., 2007) or incorporating biologically relevant handcrafted features (Guo et al., 2008; Hamp & Rost, 2015). Although they achieve notable accuracy and efficiency, the reliance on the single data modality (sequence) and the design bottleneck of models limit the expressiveness of protein representations. Recently, the rapid development of sequence modeling techniques (e.g., RCNN (Chen et al., 2019), transformers (Lin et al., 2023), and LSTM (Tsukiyama et al., 2021)) and geometric learning (Ganea et al., 2021; Wang et al., 2023; Gao et al., 2023a) has significantly enhanced the representation of protein sequences and structures. Moreover, multi-modal data representation (Wu et al., 2024; Wang et al., 2022; Gao et al., 2024a;b;c) has also demonstrated superiority in performance. However, these methods overlook how to appropriately handle these powerful representations in the classification head, leading to performance bottlenecks.

**Graph Prompt Learning (GPL).** Broadly, GPL applies learnable modifications to graph structures or attributes to enable efficient knowledge transfer and knowledge integration (Zi et al., 2024). Current approaches fall into two categories: **prompt as graph** (Sun et al., 2023; Huang et al., 2023) modifies original graph structures/nodes or inserts additional graphs via learnable manners; **prompt as vector** (Fang et al., 2023; Liu et al., 2023; Sun et al., 2022): augments node attributes with learnable vectors while preserving the original graph structure. Although existing GPL methods can effectively narrow down the gaps between different tasks and achieve performance improvements, their acquisition of new knowledge is implicit. For example, in All-in-one (Sun et al., 2023), the learned insert patterns are not interpretable and lack real-world significance. In this paper, we leverage the capability of GPL to acquire external knowledge and follow the well-known L3 rule to construct interpretable graph prompts, explicitly introducing the *complementarity* prior.

## 3. Preliminaries

In this section, **(1)** we first introduce the definitions of two mainstream protein-protein interaction (PPI) prediction scenarios. Then, **(2)** we reinforce the L3 rule and validate it using popular deep learning datasets for these scenarios.

### 3.1. Problem Definitions

We define two fundamental PPI prediction tasks: **binary PPI prediction** determines whether a given protein pair interacts, while **interaction type prediction** identifies specific interaction type(s) when an interaction is known to exist. Formal definitions are provided below:

**Scenario 1: Binary PPI Prediction.** Given a set of $N$ proteins $\mathcal{P} = \{P_1, P_2, \ldots, P_N\}$, we consider all possible pairwise combinations $\mathcal{C} = \{(P_i, P_j) \mid P_i, P_j \in \mathcal{P}, i \neq j\}$. The binary interaction status of each pair is defined as $y_{ij} \in \{0, 1\}$, where $y_{ij} = 1$ indicates interaction and $y_{ij} = 0$ indicates no interaction. We construct a PPI graph $\mathcal{G}_T = (\mathcal{V}_T, \mathcal{E}_T)$ where nodes $\mathcal{V}_T$ represent proteins and edges $\mathcal{E}_T \subseteq \mathcal{C}$ correspond to observed interacting pairs ($y_{ij} = 1$). Under semi-supervision, only a subset $\mathcal{C}_L \subseteq \mathcal{C}$ has known interaction status $\mathcal{Y}_L = \{y_{ij}\}$, forming labeled data $\mathcal{D}_L = (\mathcal{C}_L, \mathcal{Y}_L)$. The remaining pairs $\mathcal{C}_U = \mathcal{C} \setminus \mathcal{C}_L$ with unknown status constitute unlabeled data $\mathcal{D}_U$. Our objective is to learn a mapping $\mathcal{F} : \mathcal{C} \to \{0, 1\}$ using $\mathcal{D}_L$ in an *inductive* setting, enabling accurate binary prediction of interaction existence for $\mathcal{C}_U$.

**Scenario 2: Interaction Type Prediction.** Given a collection of $N$ proteins $\mathcal{P} = \{P_1, P_2, \ldots, P_N\}$ and their pairwise interactions $\mathcal{X} = \{(P_i, P_j) \mid P_i, P_j \in \mathcal{P}, i \neq j\}$, we construct a PPI graph $\mathcal{G}_T = (\mathcal{V}_T, \mathcal{E}_T)$. In this representation, nodes $\mathcal{V}_T$ correspond to proteins, while edges $\mathcal{E}_T$ encode observed interactions from $\mathcal{X}$. The task involves predicting interaction types from a predefined label space $\mathcal{L} = \{l_0, l_1, \ldots, l_n\}$ containing $n$ possible interaction types. Under semi-supervision, only a subset $\mathcal{X}_L \subseteq \mathcal{X}$ has known labels $\mathcal{Y}_L \subseteq \mathcal{L}$, forming labeled data $\mathcal{D}_L = (\mathcal{X}_L, \mathcal{Y}_L)$. The remaining interactions $\mathcal{X}_U = \mathcal{X} \setminus \mathcal{X}_L$ with unknown labels $\mathcal{Y}_U$ constitute unlabeled data $\mathcal{D}_U$. Our objective is to learn a mapping $\mathcal{F} : \mathcal{X} \to \mathcal{L}$ using $\mathcal{D}_L$ in a *transductive* setting, enabling accurate prediction of $\mathcal{Y}_U$.

### 3.2. Consolidation of the L3 Rule

Here, we provide additional empirical evidence supporting the L3 rule. Compared to (Kovács et al., 2019), we also consider Pearson Correlation and Mutual Information to evaluate the significance of L3 and introduce more competitors, such as L4 and L5. Moreover, we test two datasets: the Yeast dataset for **Scenario 1**, and SHS27k for **Scenario 2**.

Figure 2 visualizes the results. First, we confirm a finding from (Kovács et al., 2019): the principle that similarity (indicated by #L2 Paths) drives positive links does not hold for the PPI problem. Furthermore, in both datasets, #L3 Paths exhibits the strongest correlation with the presence of PPI. Interestingly, the #L5 Paths and #L7 Paths indicators demonstrate substantially stronger correlations than #L4 Paths and #L6 Paths. This phenomenon arises because the L5 path and L7 path are both structural extensions of the L3 one. For instance, consider an L3 path with complementary interaction: $a_\sqcup \to b_\sqcap \to c_\sqcup \to d_\sqcap$, where $\sqcup/\sqcap$ denote concave/convex domains. Inserting an additional concave-convex pair between proteins $b_\sqcap$ and $c_\sqcup$, it becomes an L5 path that preserves complementarity principles: $a_\sqcup \to b_\sqcap \to x_\sqcup \to y_\sqcap \to c_\sqcup \to d_\sqcap$.

## 4. Proposed Method

**Overview.** We propose L3-PPI, an L3-path-regularized graph prompt learning framework (Figure 3) that enhances existing PPI predictors by incorporating complementarity priors. The core idea is to generate the L3 pattern for a pair of proteins while constraining the number of active L3 paths according to PPI presence: **fewer activated L3 paths for non-interacting pairs, more for interacting pairs.**

The framework operates through three sequential stages: (1) Pre-training an L3 pattern recognition surrogate model to validate L3 pattern graphs; (2) Constructing an **initial** L3 pattern with $K$ paths featuring learnable node attributes (excluding query nodes), then filtering paths through a gating network; (3) Combining selected paths into the **final** prompt pattern, which is processed by the pre-trained surrogate.

A key regularization mechanism controls path retention by modulating the gating network's output probabilities, ensuring path counts align with interaction labels. The surrogate's classification output serves as the PPI likelihood.

### 4.1. L3 Pattern Recognition Pre-training

Within the pre-training and prompt tuning framework, pre-training is essential for knowledge acquisition and generalization. Crucially, the pre-training and prompt tuning tasks must occupy a shared intrinsic task subspace.

In this work, we aim to optimize the number of L3 paths (#L3 paths) during prompt tuning to align with the L3 rule. However, the virtually generated L3 patterns may not necessarily meet the data distribution in the original dataset, potentially compromising generalization. To address this, we formulate the pre-training task as L3 pattern graph validity prediction, enabling explicit learning of native L3 pattern distributions to guide prompt optimization.

**Definition 4.1** (Pre-training Dataset $\mathcal{D}_{pre}$). Given a PPI network $G = (V, E)$ where $V$ is the protein set and $E$

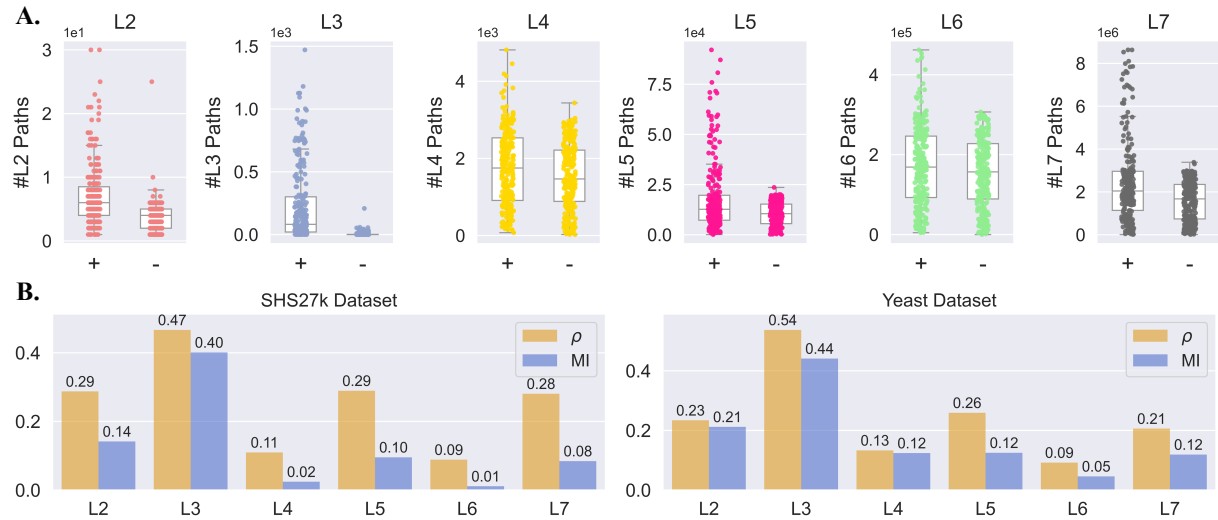

*Figure 2.* Empirical evidences supporting the L3 rule. (A). We compute the number of L$i$ paths (#L$i$ Paths, $i = 2$–$7$) for 500 positive ('+') and 500 negative ('-') links sampled from the SHS27k dataset. (B). We report the Pearson correlation coefficient ($\rho$) and mutual information (MI) between the number of L$i$ paths ($i = 2$–$7$) and link labels (0/1) across two datasets.

represents interactions, we compute node embeddings $e_v \in \mathbb{R}^d$ for all $v \in V$ using a PPI predictor (e.g., PIPR (Chen et al., 2019)). The **positive samples** $\mathcal{D}_{pre}^+$ consist of all L3 paths between interacting pairs $(u, v) \in E$, obtained via depth-first search. **Negative samples** $\mathcal{D}_{pre}^-$ comprise all L3 paths between non-interacting pairs $(u, v) \notin E$. The final dataset is $\mathcal{D}_{pre} = \{(\mathcal{G}_{pre}, y_{pre})\}$ where $\mathcal{G}_{pre}$ denotes a pattern graph and $y_{pre} \in \{0, 1\}$ the label.

We apply Graph Neural Networks (GNNs) (Kipf, 2016; Xu et al., 2018; Hamilton et al., 2017; Velickovic et al., 2017) for graph-level binary classification. In the pre-training phase, we apply Graph Isomorphism Network (GIN (Xu et al., 2018)) along with a classification head to form a GNN model that classifies the input graph $\mathcal{G}_{pre}$. The pre-training model approximates the L3 pattern validity with data instances of $\mathcal{D}_{pre}$ defined in Defi 4.1:

$$\tilde{y}_{pre} = \text{GNN}_{pre}(\mathcal{G}_{pre}; \theta, \phi) \approx y_{pre}, \tag{1}$$

where $\text{GNN}_{pre}$ represents a combination of a GIN with parameters $\theta$ for obtaining node embeddings, a ReadOut function following the last GIN layer, and a task head with parameters $\phi$ to produce the prediction $\tilde{y}_{\text{pre}}$.

To summarize, the complete optimization of the pre-training process is as follows:

$$\theta^*, \phi^* =$$
$$\underset{\theta, \phi}{\arg\min} \sum_{(\mathcal{G}_{pre}, y_{pre}) \in \mathcal{D}_{pre}} \mathcal{L}_{pre}\Big(y_{pre}, \text{GNN}_{pre}(\mathcal{G}_{pre}; \theta, \phi)\Big), \tag{2}$$

where $\mathcal{L}_{pre}$ is the binary cross entropy loss function.

### 4.2. Graph Prompt Tuning on PPI Prediction

**Definition 4.2** (Prompt Tuning Dataset $\mathcal{D}_{gpt}$). Given a PPI network with a set of $N$ proteins $\mathcal{P} = \{P_1, P_2, \ldots, P_N\}$, we consider the pairwise embedding combination $e_{gpt} = (\mathcal{M}(P_i), \mathcal{M}(P_j))$, where $P_i, P_j \in \mathcal{P}, i \neq j$ and $\mathcal{M}$ represents any existing PPI prediction method that we aim to integrate. The final dataset is $\mathcal{D}_{gpt} = \{e_{gpt}, y_{gpt}\}$, where $e_{gpt}$ denotes an embedding pair and $y_{gpt} \in \{0, 1\}$ the label.

#### 4.2.1. PROMPT DESIGN

First, we define the dataset for the graph prompt tuning (GPT), as shown in Defi. 4.2. For clarity, we denote each data instance as a tuple form $(u, v, y_{gpt}) \in \mathcal{D}_{gpt}$, where $u, v$ denote the query nodes (proteins) with attributes $e_{gpt}[0] \in \mathbb{R}^d$ and $e_{gpt}[1] \in \mathbb{R}^d$, where $d$ represents the latent dimensionality of the predictor $\mathcal{M}$. Our prompt design shown in Figure 4 consists of three parts: prompt nodes, prompt structure and inserting manner. We define the set of $K + 1$ **prompting nodes** $V^P = \{v_0^P, v_1^P, ..., v_K^P\}$ that simulates virtual proteins. Each node has a embedding vector denoted by $x_i \in \mathbb{R}^d$ for node $v_i^P$. Thus, we have the set of learnable prompt embeddings denoted as $X^P = \{x_0^P, x_1^P, ..., x_K^P\}$. Considering all $K + 1$ prompt nodes, our **prompt structure** is represented by the edge set:

$$E^P = \{(v_0^P, v_1^P), (v_0^P, v_2^P), ..., (v_0^P, v_K^P)\}. \tag{3}$$

The inserting manner is typically also represented by a set of edges, indicating the way prompt nodes are inserted into query nodes. We can define our **inserting manner** as:

$$E^I = \{(v_0^P, v), (v_1^P, u), (v_2^P, u), ..., (v_K^P, u)\} \tag{4}$$

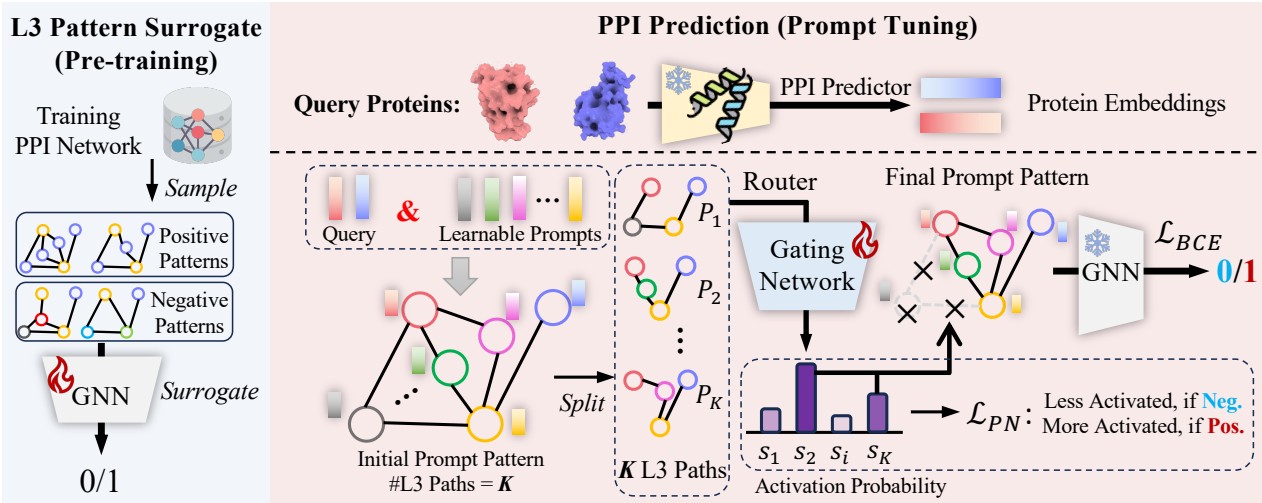

Figure 3. **The method overview of proposed L3-PPI.** Our method operates in a plug-and-play manner, enabling lightweight integration as a classification head after any leading PPI predictor while keeping the predictor frozen. The whole framework contains a pre-trained Graph Neural Networks (GNNs) serving as a surrogate for L3 pattern recognition, learnable prompts representing virtual proteins within the initial prompt L3 pattern, and a gating network to select activated L3 paths. The PPI prediction results are obtained after the final prompt pattern passes through the pre-trained surrogate. The $\mathcal{L}_{PN}$ loss, which controls the #L3 paths and the classification loss $\mathcal{L}_{BCE}$ jointly optimize the model.

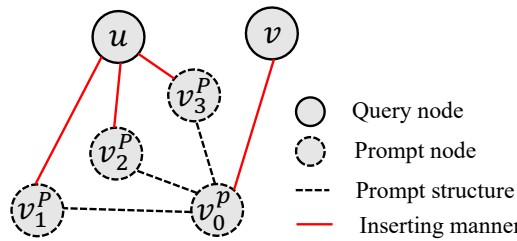

Figure 4. We present an example with $K = 3$. Query nodes $(u, v)$, prompt nodes $(v_0^P, v_1^P, v_2^P, v_3^P)$, prompt structures (black dashed lines), and inserting manner (red solid lines) form an L3 pattern that contains three L3 paths.

The structure is fixed. However, the vectorized embeddings of the $K + 1$ prompt nodes are learnable and shared across all query pairs, where $K$ is a hyperparameter.

#### 4.2.2. GATING NETWORKS FOR FILTERING L3 PATHS

During prompting, 2 query nodes and $K + 1$ prompt nodes collectively simulate an L3 pattern in a *virtual* PPI network. This **initial prompt pattern** remains largely consistent across different query pairs, differing in the embeddings of query nodes $u, v$. To guarantee distinctiveness across prompt patterns for different query pairs, we introduce a sparsely-gated mechanism that can select appropriate L3 paths for each query node pair from $K$ candidate paths.

We decompose the initial prompt pattern into $K$ L3 paths, denoted as $\{path_k\}_{k=1}^K$, where each path $path_i$ is

a graph with the node set: $\{u, v, v_0^P, v_i^P\}$ and edge set $\{(u, v_i^P), (v_i^P, v_0^P), (v_i^P, v)\}$.

Since binarization methods such as ArgMax and Bernoulli sampling typically lack differentiability, we employ Gumbel-Softmax reparameterization to train the gating network. The gating function $g$ for path $path_i$ outputs:

$$g(path_i) = \text{Sigmoid}\left(\frac{\log p_i + \epsilon - \log(1 - p_i) - \epsilon'}{\tau}\right)$$
(5)

where $p_i = \text{GNN}_{gpt}(path_i)$ denotes the gate activation probability, and $\epsilon, \epsilon' \overset{\text{i.i.d.}}{=} -\log(-\log(u))$ with $u \sim \mathcal{U}(0, 1)$ are independent Gumbel noise variables. The temperature hyperparameter $\tau$ is annealed during training. During backpropagation, we use $g(path_i)$; at inference, we threshold $p_i$ to obtain the binary activation $\mathbb{I}_{[p_i > 0.5]}$.

#### 4.2.3. REFORMULATING THE PPI PREDICTION TASK

A key aspect of prompt learning is how to reformulate prompt tuning into the pre-training task to ensure consistency between two spaces. We therefore transform pairwise protein prediction into graph-level prompt pattern prediction, aligning with the pre-training established in Section 4.1.

We remove the non-activated L3 paths through zero-weight assignment to their edges. Specifically, we assign the binary gating outputs from Equation 5 as weights of edges that are in corresponding L3 paths.

$$\{w(u, v_i^P), w(v_i^P, v_0^P), w(v_i^P, v)\} \leftarrow g(path_i), \quad (6)$$

for $i = 1, 2, ..., K$, where $w(u, v)$ represents the weight of edge between nodes $u$ and $v$.

Finally, we input the **final prompt pattern** $\mathcal{G}_F$, composed of $K + 3$ nodes, $2K + 1$ edges, and edge weights, into the pre-trained surrogate $\text{GNN}_{pre}$ for PPI prediction results:

$$\tilde{y}_{gpt} = \text{GNN}_{pre}(\mathcal{G}_F; \theta^*, \phi^*) \approx y_{gpt}, \tag{7}$$

where $\tilde{y}_{gpt}$ represents the final PPI prediction result and the pre-trained $\text{GNN}_{pre}$ parameterized by $\theta^*, \phi^*$ is frozen.

### 4.3. Optimization

During prompt tuning, the primary loss function is the binary cross entropy (BCE) between predictions and ground-truth labels: $\mathcal{L}_{BCE} = \text{BCE}(\tilde{y}_{gpt}, y_{gpt})$. Moreover, we encourage the model to retain more activated L3 paths for positive samples and fewer for negative samples. This is achieved with an additional path number regularizing loss:

$$\mathcal{L}_{PN} = \begin{cases} \max\left(0, \; K\left(1 - \frac{1}{\gamma}\right) - \sum_{i=1}^{K} p_i\right) & \text{if } y_{gpt} = 1 \\ \max\left(0, \; \sum_{i=1}^{K} p_i - \frac{K}{\gamma}\right) & \text{if } y_{gpt} = 0 \end{cases} \tag{8}$$

Given the need to optimize two distinct parameter sets during prompt tuning (i.e., prompt embeddings and the gating network), we adopt a two-stage training strategy to ensure stable optimization. In the first stage, we update the prompt embeddings $X^P = \{x_0^P, x_1^P, ..., x_K^P\}$ using $\mathcal{L}_{BCE}$. In the second stage, we jointly optimize all parameters including both parameter $X^P$ and the parameter of the gating network by using the combined loss $\mathcal{L}_{BCE} + \mathcal{L}_{PN}$.

## 5. Experiments

### 5.1. Dataset

We evaluate our proposed model on multiple PPI datasets covering both interaction type prediction and binary PPI prediction tasks. For the interaction type prediction setting, we follow (Chen et al., 2019; Wu et al., 2024; Lv et al., 2021) and adopt three benchmarks from the Homo sapiens subset of the STRING database (Szklarczyk et al., 2023), which integrates and scores publicly available PPI information to construct a comprehensive global network including both direct (physical) and indirect (functional) interactions. Specifically, we employ the **STRING** dataset comprising 14,952 proteins and 572,568 PPIs across seven distinct interaction types (activation, inhibition, catalysis, expression, and three others). We further evaluate on two challenging subsets: **SHS27k** (16,912 PPIs) and **SHS148k** (99,782 PPIs), both containing proteins with $> 50$ amino acids and $< 40\%$ sequence identity. For the binary PPI prediction task, we construct a dataset by extracting from

**SHS27k** only those protein pairs annotated with **binding** interactions, treating protein pairs without binding evidence as non-interacting. We also include the widely used **Yeast** PPI dataset to provide complementary evaluation across species. Following (Wu et al., 2024; Lv et al., 2021), we further adopt Random, Breadth-First Search (BFS), and Depth-First Search (DFS) strategies for dataset splitting. These more challenging splitting methods (BFS and DFS) partition the test data into three subsets: BS (both proteins seen in training), ES (either protein seen), and NS (neither protein seen). To our knowledge, no prior work has established BFS or DFS data partitioning schemes for the task of binary PPI prediction. We bridge this research gap, with details provided in the Appendix. Following (Wu et al., 2024), we partition the PPI data into training (60%), validation (20%), and test (20%) sets across all baselines, employing three distinct splitting methods: Random, BFS, and DFS. Moreover, since the results of BFS and DFS vary with the choice of root nodes, we restrict the root degree to $\leq 5$ to better reflect the realistic scenario.

### 5.2. Evaluation Metrics.

We adopt micro-F1 for multi-label interaction type prediction evaluation due to its sample-wise averaging, which: (1) handles class imbalance by equal sample weighting, and (2) better reflects performance on prevalent interaction types in our imbalanced datasets. Then, we select the model that performs the best on the validation set to evaluate the micro-F1 scores of the test data. Besides, to reduce randomness from data partitioning, we repeat each experiment with three different seeds and report the averaged performance.

### 5.3. Baselines

We evaluate L3-PPI from two perspectives. (1). We test its ability to boost existing DL methods. DPPI (Hashemifar et al., 2018), SemiGNN-PPI (Zhao et al., 2023), DNN-PPI (Li et al., 2018), S2F (Xue et al., 2022), PIPR (Chen et al., 2019), GNN-PPI (Lv et al., 2021), HIGH-PPI (Gao et al., 2023a), and MAPE-PPI (Wu et al., 2024). (2). We examine its adaptability to boost protein pre-training models: ProstT5 (Heinzinger et al., 2024), Saprot (Su et al., 2023), ESM2 (Lin et al., 2022), and GearNet (Zhang et al., 2022).

### 5.4. Results

Tables 1 and 2 present model performance on interaction type prediction (ITP) and binary PPI prediction (BPP), respectively. Our method improves performance across both tasks compared to baseline methods. Notably, L3-PPI achieves an average improvement of up to 3.88 based on PIPR across 9 scenarios (3 datasets × 3 partitions). A key finding is the strong adaptability of our L3-PPI classification head to protein pre-training models. Combining these

*Table 1.* **Performance comparison on the interaction type prediction task.** The "+" suffix in **Method** indicates integration into our L3-PPI framework. ▲/▼ indicates better/worse than without L3-PPI. Best results in each partition are **bold**.

| Method | SHS27k | | | SHS148k | | | STRING | | | Avg. Gain |
|---|---|---|---|---|---|---|---|---|---|---|
| | Random | DFS | BFS | Random | DFS | BFS | Random | DFS | BFS | |
| DPPI | 70.45 | 43.69 | 43.87 | 76.10 | 51.43 | 50.80 | 92.49 | 63.41 | 54.41 | - |
| DPPI+ | 75.62 ▲ | 46.79 ▲ | 47.46 ▲ | 79.37 ▲ | 54.65 ▲ | 55.28 ▲ | 92.99 ▲ | 66.93 ▲ | 55.03 ▲ | +3.05 |
| SemiGNN-PPI | 85.57 | 69.25 | 67.94 | 91.40 | 77.62 | 71.06 | 94.80 | 84.85 | 77.10 | - |
| SemiGNN-PPI+ | 83.21 ▼ | 77.49 ▲ | 71.92 ▲ | 91.69 ▲ | 79.48 ▲ | **76.40** ▲ | 95.26 ▲ | 84.66 ▼ | 79.81 ▲ | +2.26 |
| DNN-PPI | 75.18 | 48.90 | 51.59 | 85.44 | 56.70 | 54.56 | 81.91 | 61.34 | 51.53 | - |
| DNN-PPI+ | 79.39 ▲ | 52.96 ▲ | 51.97 ▲ | 89.03 ▲ | 62.32 ▲ | 57.92 ▲ | 84.97 ▲ | 65.39 ▲ | 52.66 ▲ | +3.27 |
| S2F | 73.71 | 44.68 | 46.32 | 80.67 | 56.06 | 50.25 | 85.46 | 55.07 | 62.31 | - |
| S2F+ | 75.60 ▲ | 46.60 ▲ | 49.03 ▲ | 84.35 ▲ | 57.03 ▲ | 57.68 ▲ | 87.36 ▲ | 58.96 ▲ | 62.71 ▲ | +2.75 |
| PIPR | 79.59 | 52.19 | 47.13 | 88.81 | 61.38 | 58.57 | 93.68 | 64.97 | 53.80 | - |
| PIPR+ | 83.22 ▲ | 57.97 ▲ | 49.02 ▲ | 88.98 ▲ | 64.52 ▲ | 66.37 ▲ | 94.30 ▲ | 69.33 ▲ | 61.32 ▲ | +3.88 |
| GNN-PPI | 83.65 | 66.52 | 63.08 | 90.87 | 75.34 | 69.53 | 94.53 | 84.28 | 75.69 | - |
| GNN-PPI+ | 87.42 ▲ | 67.78 ▲ | 69.52 ▲ | 92.23 ▲ | 79.58 ▲ | 77.46 ▲ | 95.29 ▲ | 86.32 ▲ | **79.91** ▲ | +3.56 |
| HIGH-PPI | 86.23 | 70.24 | 68.40 | 91.26 | 78.18 | 72.87 | - | - | - | - |
| HIGH-PPI+ | 86.47 ▲ | 72.57 ▲ | 73.22 ▲ | 93.45 ▲ | **82.80** ▲ | 76.03 ▲ | - | - | - | +2.89 |
| MAPE-PPI | **88.91** | 71.98 | 70.38 | 92.38 | 79.45 | 74.76 | 96.12 | 86.50 | 78.26 | - |
| MAPE-PPI+ | 87.93 ▼ | 74.69 ▲ | **74.50** ▲ | 93.41 ▲ | 79.99 ▲ | 75.42 ▲ | 95.71 ▼ | 87.66 ▲ | 79.32 ▲ | +1.10 |
| ProstT5+ | 88.37 | 75.81 | 67.42 | 90.31 | 77.83 | 68.92 | **96.76** | 84.25 | 72.66 | - |
| ESM2-650M+ | 85.16 | 68.43 | 71.25 | 91.90 | 74.67 | 73.46 | 95.56 | 84.17 | **79.91** | - |
| Saprot+ | 83.21 | 73.44 | 71.51 | **93.66** | 80.09 | 74.63 | 93.96 | 84.49 | 78.32 | - |
| GearNet+ | 87.18 | 73.27 | 65.33 | 89.46 | 81.29 | 71.20 | 93.96 | 83.25 | 79.03 | - |

achieves competitive results, yielding the best performance in 3 out of 9 ITP scenarios and 3 out of 6 BPP scenarios.

While our method shows limited improvement and occasionally slight degradation on Random partitioning, it delivers significant enhancements for nearly all baselines under BFS and DFS partitioning. This observation aligns with our core motivation: 1) In generalization-focused scenarios (BFS/DFS), the complementarity prior introduced by L3-PPI effectively strengthens the model; 2) These partitioning schemes inherently create disconnected test sets (and sometimes training sets). Crucially, L3-PPI operates without dependency on connected network structures, processing independent PPI pairs directly.

Moreover, we observe that for certain baselines with inherently low performance (such as SemiGNN-PPI), our improvements may not be significant. This limitation arises because L3-PPI functions primarily as a classification head designed to leverage strong representations effectively. When the input representations themselves lack expressiveness, our method may struggle to deliver substantial gains.

### 5.5. Generalization Ability

We show in Figure 5 the specific performance of L3-PPI on the BS, ES, and NS sample categories, including performance (A) and inference capability (B). Notably, L3-PPI achieves performance improvements on all categories of samples, showing particularly robust enhancement in the more challenging ES and NS categories. In addition, we find that the designed regularization term $\mathcal{L}_{PN}$ is effective, as it accurately recovers the distribution of the actual #L3 paths

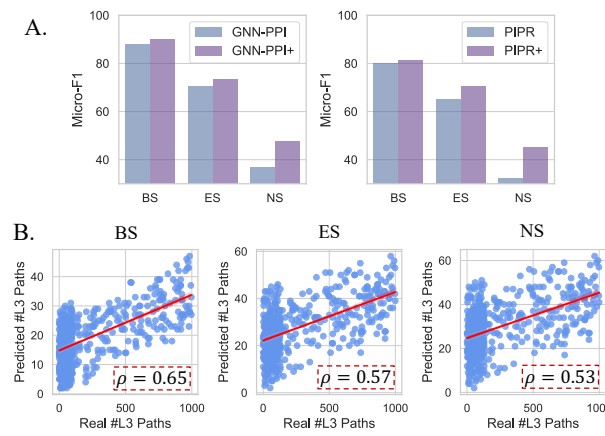

*Figure 5.* (A) Performance comparison of our method *vs.* 2 baselines on BS, ES and NS categories (Random partition). (B) We set the upper limit for the number of prompt L3 paths, denoted as $K$, to 50, and we demonstrate the relationship between the number of **inferred** (predicted) #L3 paths and the **actual** #L3 paths in the PPI network. $\rho$ refers to the Pearson correlation coefficient.

for all three categories. Since #L3 paths is a strong indicator of PPI, accurately inferring this intermediate variable can help the model enhance its generalization.

### 5.6. Ablation Study

We conduct an ablation study on two elements: the maximum number of prompt L3 paths ($K$) and the L3-PPI model components, notably the L3 regularization loss ($\mathcal{L}_{PN}$).

*Table 2.* **Performance comparison on the binary PPI prediction task.** The "+" suffix in **Method** indicates integration into our L3-PPI framework. ▲/▼ indicates better/worse than without L3-PPI. Best results in each partition are **bold**.

| Method | SHS27k (Binding) | | | Yeast | | | Avg. Gain |
| --- | --- | --- | --- | --- | --- | --- | --- |
| | Random | DFS | BFS | Random | DFS | BFS | |
| DPPI | 65.17 | 50.98 | 55.72 | 74.69 | 61.60 | 60.57 | - |
| DPPI+ | 69.94 ▲ | 53.22 ▲ | 57.01 ▲ | 77.28 ▲ | 63.03 ▲ | 61.54 ▲ | +2.22 |
| MLP | 76.54 | 50.12 | 56.77 | 90.16 | 59.62 | 55.54 | - |
| MLP+ | 75.43 ▼ | 54.92 ▲ | 63.70 ▲ | 92.53 ▲ | 65.72 ▲ | 62.73 ▲ | +4.38 |
| DNN-PPI | 63.62 | 44.20 | 49.05 | 89.77 | 61.89 | 60.94 | - |
| DNN-PPI+ | 67.71 ▲ | 44.23 ▲ | 51.71 ▲ | 90.61 ▲ | 62.24 ▲ | 64.49 ▲ | +1.92 |
| PIPR | 79.53 | 59.60 | 52.42 | 93.92 | 67.17 | 60.96 | - |
| PIPR+ | **81.27** ▲ | 63.79 ▲ | 54.30 ▲ | **94.55** ▲ | 69.18 ▲ | 65.66 ▲ | +2.53 |
| GNN-PPI | 77.78 | 45.97 | 58.04 | 91.42 | 66.78 | 63.24 | - |
| GNN-PPI+ | 80.02 ▲ | **66.09** ▲ | 67.01 ▲ | 93.57 ▲ | 66.66 ▼ | 65.97 ▲ | +6.02 |
| ProstT5+ | 77.21 | 59.64 | 56.31 | 94.33 | 65.96 | 66.71 | - |
| ESM2-650M+ | 79.62 | 63.77 | 65.09 | 93.05 | **71.94** | **70.56** | - |
| Saprot+ | 73.33 | 64.60 | **67.23** | 93.49 | 70.39 | 70.72 | - |
| GearNet+ | 80.79 | 65.73 | 59.60 | 91.66 | 69.90 | 63.22 | - |

**Model Components.** We can observe from the results in Table 4 that (1) the regularization and the gating network jointly play an important role. More specifically, even without $\mathcal{L}_{PN}$, relatively satisfactory results can be achieved because the gating mechanism can still help learn part of the L3 rules. However, once the gating network is removed, the model will completely fail. (2) The performance drop resulting from the removal of pre-training underscores that the process learns transferable knowledge crucial for the downstream PPI prediction task. (3) The irreplaceability of GIN in the L3-PPI classification head lies in its superior capacity to capture the nuanced structural and feature differences across diverse prompt L3 patterns.

**The $K$ Value.** As shown in Figure 6, the value of $K$ is a key parameter that influences prediction performance. Nonetheless, the distribution of the optimal $K$ values exhibits a clear trend. A clear increasing trend in the optimal $K$ value is observed across datasets SHS27k, SHS148k, and STRING, which correlates with their increasing scales. This finding holds regardless of the base model, indicating that we can efficiently search for this hyper-parameter within a small range when enhancing any PPI predictor.

*Table 3.* Ablation study results (Micro-F1) on three benchmarks with PIPR+ (PIPR+L3-PPI). Best results are **bold**. We perform hyperparameter searches individually for each ablation model, rather than reusing the optimal set from the "full model".

| Model | SHS27k | SHS148k | STRING |
| --- | --- | --- | --- |
| Full model | **83.22** | **88.08** | **94.30** |
| without regularization $\mathcal{L}_{PN}$ | 80.10 | 85.17 | 90.92 |
| without gating network $g$ | 76.56 | 79.90 | 87.17 |
| without pre-training | 81.92 | 86.77 | 91.91 |
| replacing GIN with GCN | 83.12 | 87.97 | 91.45 |
| replacing GIN with GAT | 82.72 | 86.95 | 94.17 |

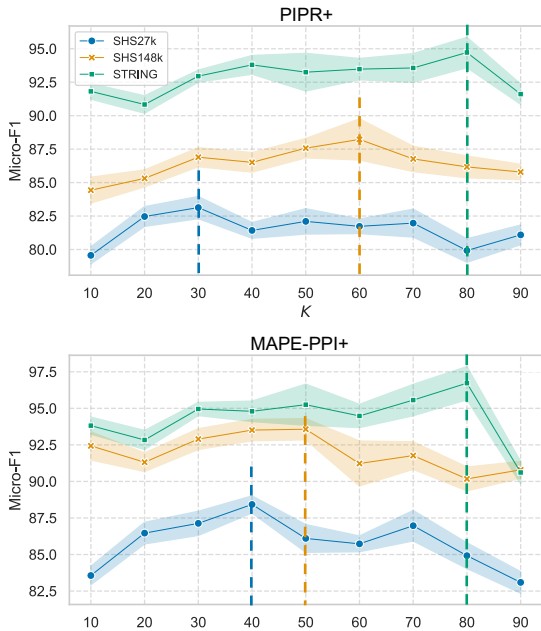

*Figure 6.* The trend of performance with respect to changes in $K$.

## 6. Conclusion

We propose **L3-PPI**, a model-agnostic framework that incorporates biologically grounded complementarity priors into protein-protein interaction (PPI) prediction through *L3-path-regularized graph prompt learning*. Our method generates virtual L3 paths and dynamically regulates path counts based on interaction likelihood, which is consistent with the L3 rule. Unlike existing graph prompting methods, the proposed approach constrains the generated graph structures to possess specific patterns, resulting in greater interpretability. This lightweight, plug-and-play classification head shows broad compatibility with diverse PPI predictors.

## Impact Statement

This paper presents work whose goal is to advance the field of applications in Machine Learning. There are many potential societal consequences of our work, none which we feel must be specifically highlighted here.

## Acknowledgments and Disclosure of Funding

This work is supported by NSFC 62572418, Guangdong Provincial Talent Program (No. 2024TQ08X366), Shenzhen Hetao Shenzhen-Hong Kong Science and Technology Innovation Cooperation Zone (No. HTHZQSWS-KCCYB-2023052).

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

## Technical Appendix

In the technical appendix, we present the data processing techniques, detailed implementations of the experiments, and additional experimental results.

This appendix is outlined as follows:

- Section A provides a detailed introduction to the dataset, including statistical information and collection methods. We also describe the BFS and DFS data partitioning methods for binary PPI prediction.

- Section B provides additional experimental results, including computational efficiency, generalization ability *etc*.

- Section C provides detailed implementation details of the experiments, including hyperparameters choices and training techniques.

## A. Dataset and Data Partitioning

### A.1. Dataset

First, we considered two types of PPI prediction tasks: interaction type prediction and binary PPI prediction. The former involves the SHS27k, SHS148k, and STRING datasets, while the latter used a subset of the SHS27k data and the whole Yeast dataset. For interaction type prediction, the datasets used in this study were sourced exclusively from the STRING database (Szklarczyk et al., 2023). This database compiles protein-protein interaction (PPI) data from most publicly available sources, applies scoring metrics, and integrates the information to construct a comprehensive, objective global PPI network.

The largest dataset, named the STRING dataset (hereafter, "STRING" refers specifically to this dataset, not the database), includes seven interaction types: direct (physical) and indirect (functional) interactions categorized as reaction, binding, post-translational modifications (ptmod), activation, inhibition, catalysis, and expression. STRING contains 15,335 proteins and 593,397 PPIs. Critically, every interaction between a protein pair is supported by at least one of these seven link types.

The other two datasets, SHS27k and SHS148k, are Homo sapiens-specific subsets sampled from the STRING dataset by (Chen et al., 2019):

- SHS27k: 1,690 proteins and 7,624 PPIs

- SHS148k: 5,189 proteins and 44,488 PPIs

For binary PPI prediction, we consider Yeast and SHS27k (binding) datasets. The Yeast dataset, belonging to Guo's dataset (Guo et al., 2008) collection, is a widely adopted benchmark. It comprises 2,497 proteins forming 11,188 PPIs, equally split into positive and negative cases. Positive cases are derived from the Database of Interacting Proteins (DIP) (Salwinski et al., 2004), excluding proteins with fewer than 50 amino acids or $\geq$40% sequence identity. Full protein sequences are sourced from UniProt (UniProt Consortium, 2018). Negative cases are generated by randomly pairing proteins without known interactions, excluding pairs sharing the same sub-cellular location.

Additionally, to ensure a more comprehensive evaluation, we extracted the binding type data from SHS27k and treated it as an independent dataset. Therefore, it is an imbalanced link prediction dataset.

### A.2. Data Partitioning

As we mentioned before, we define two fundamental PPI prediction tasks: **binary PPI prediction** (BPP) determines whether a given protein pair interacts, while **interaction type prediction** (ITP) identifies specific interaction type(s) when an interaction is known to exist. For **interaction type prediction**, we follow (Lv et al., 2021) to employ Random, Breadth-First Search (BFS), and Depth-First Search (DFS) strategies for dataset splitting. However, for BPP the test links are not all interacting proteins like those in ITP. Instead, they include both positive and negative links, suggesting that we cannot directly apply the data partitioning method from (Lv et al., 2021) to BPP.

We design two heuristic evaluation schemes based on the PPI network, namely BFS and DFS, to generate the test set for the binary classification task. Specifically, the construction of the positive and negative pairs in $\mathcal{X}_{\text{test}}$ is aligned with Algorithm 1.

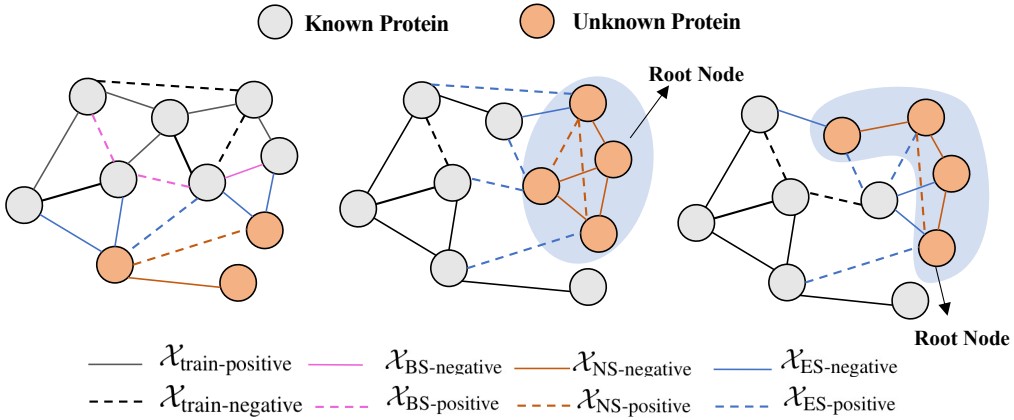

*Figure 7.* Examples of different testset construction strategies for binary PPI prediction.

Clustered scenario (BFS): To simulate the case where unknown proteins form densely connected clusters (Figure 6, we first select a root node $p_{\text{root}}$ whose degree satisfies $|\mathcal{N}(p_{\text{root}})| \geq t$. Starting from $p_{\text{root}}$, we expand via Breadth-First Search (BFS) until the number of collected positive PPIs reaches $N/2$, forming $\mathcal{X}_{\text{test-positive}}$. To balance the test set, we then sample an equal number of non-interacting pairs as $\mathcal{X}_{\text{test-negative}}$.

Sparse scenario (DFS): To simulate the case where unknown proteins are sparsely distributed with few interactions (Figure 6, we instead apply Depth-First Search (DFS) from $p_{\text{root}}$.) This procedure selects positive PPIs that are relatively scattered in the network while maintaining the connectivity of both $\mathcal{X}_{\text{train}}$ and $\mathcal{X}_{\text{test}}$. Similarly, an equal number of negatives are sampled to form $\mathcal{X}_{\text{test-negative}}$.

In both cases, $\mathcal{N}(p)$ returns the neighbors of a protein $p$, and the procedure Search_Next determines the next node according to the chosen search strategy. The final training and test sets are then constructed as

$$\mathcal{X}_{\text{train}} = \mathcal{X}_{\text{train-positive}} \cup \mathcal{X}_{\text{train-negative}}, \tag{9}$$
$$\mathcal{X}_{\text{test}} = \mathcal{X}_{\text{test-positive}} \cup \mathcal{X}_{\text{test-negative}}. \tag{10}$$

---

**Algorithm 1** Data Partition Algorithm

---

**Require:** Protein set $\mathcal{P}$; PPI set $\mathcal{X}$; Testset size $N$; Root node selection threshold $t$; Search order $\mathcal{S} \in \{\text{BFS}, \text{DFS}\}$
**Ensure:** $\mathcal{X}_{\text{train}}, \mathcal{X}_{\text{test}}$
 1: Build PPI graph $\mathcal{G} = (\mathcal{P}, \mathcal{X})$
 2: **repeat**
 3:      Randomly select a protein as root node $p_{\text{root}}$
 4: **until** $|\mathcal{N}(p_{\text{root}})| < t$
 5: $\mathcal{X}_{\text{test-positive}} \leftarrow \emptyset,\ p_{\text{cur}} \leftarrow p_{\text{root}}$
 6: **repeat**
 7:      $\mathcal{X}_{\text{cur}} \leftarrow \{\{p_{\text{cur}}, p_k\} \mid p_k \in \mathcal{N}(p_{\text{cur}})\}$
 8:      $\mathcal{X}_{\text{test-positive}} \leftarrow \mathcal{X}_{\text{test-positive}} \cup \mathcal{X}_{\text{cur}}$
 9:      $p_{\text{cur}} \leftarrow \text{Search\_Next}(\mathcal{G}, p_{\text{cur}}, \mathcal{S})$
10: **until** $|\mathcal{X}_{\text{test-positive}}| \geq N/2$
11: $\mathcal{X}_{\text{train-positive}} \leftarrow \mathcal{X} - \mathcal{X}_{\text{test-positive}}$
12: $\mathcal{X}_{\text{test-negative}} \leftarrow \text{SampleNegatives}(\mathcal{P}, \mathcal{X}, |\mathcal{X}_{\text{test-positive}}|)$
13: $\mathcal{X}_{\text{train}} \leftarrow \mathcal{X}_{\text{train-positive}} \cup \mathcal{X}_{\text{train-negative}}$
14: $\mathcal{X}_{\text{train-negative}} \leftarrow \text{SampleNegatives}(\mathcal{P}, \mathcal{X}_{\text{test}}, |\mathcal{X}_{\text{train-positive}}|)$
15: $\mathcal{X}_{\text{test}} \leftarrow \mathcal{X}_{\text{test-positive}} \cup \mathcal{X}_{\text{test-negative}}$
16: **return** $\mathcal{X}_{\text{train}}, \mathcal{X}_{\text{test}}$

---

# B. Additional Experimental Results

### B.1. Detailed Results

In Table 5, we present a more detailed comparison of the performance of GNN-PPI and GNN-PPI+ on the BS, ES, and NS samples for each partitioning method. It is evident that our approach shows only a slight advantage on the BS subset, but demonstrates a significant advantage on the ES and NS subsets. Overall, our method outperforms GNN-PPI across all three partitioning methods on three benchmarks.

### B.2. Training Efficiency

In Table 6, we show the efficiency of the whole model training process. It is clear that our method generally outperforms the baseline, and as the data scale increases, our efficiency advantage becomes even more pronounced.

# C. Implementation Details

### C.1. Hyperparameter

We perform a hyperparameter search using the configurations listed in Table 4. For each dataset, we transfer the optimal hyperparameters identified for MAPE-PPI+ (based on its validation performance) to all other methods (e.g., GNN-PPI+ and PIPR+) for evaluation.

| Hyperparameters | Values |
| --- | --- |
| Maximum #L3 path in prompt ($K$) | $4, 16, 64$ |
| Layer number of $\text{GNN}_{pre}$ | $1, 2, 4$ |
| Dimension of $\text{GNN}_{pre}$ | $32, 64, 128$ |
| Layer number of $\text{GNN}_{gpt}$ | $1, 2, 4$ |
| Dimension of $\text{GNN}_{gpt}$ | $64, 128$ |
| $\gamma$ in Equation 8 | $1.5, 2, 3$ |
| Weight of loss $\mathcal{L}_{BCE}$ | $1.0$ |
| Weight of loss $\mathcal{L}_{NP}$ | $0.1, 0.3, 0.5, 0.7$ |
| Dropout rate | $0.1, 0.2$ |
| Batch size | $16, 64$ |
| Learning rate | $1 \times 10^{-4}, 1 \times 10^{-3}$ |
| Optimizer | Adam |

*Table 4.* Hyperparameter choices of L3-PPI.

### C.2. Training

During the prompt tuning phase, the joint optimization of the gating network and prompt embedding parameters in L3-PPI poses certain challenges for training. To alleviate convergence issues, we employ a two-stage training approach. In the first stage, we ensure that all L3 paths are present and train the prompt embeddings until the model converges. This is because retaining all L3 paths presents a challenging training scenario, as the pattern graphs of all samples are quite similar, with only two query embeddings differing. By enabling gating afterward, the prompt embeddings can be further updated on a solid foundation, facilitating model convergence.

To verify this, we perform convergence experiments under several setups. The results are shown in Table 7. The selected solution has the highest overall training efficiency and the best generalization performance.

| Dataset | Partition Scheme | BS | | ES | | NS | | Average | |
|---|---|---|---|---|---|---|---|---|---|
| | | GNN-PPI | GNN-PPI+ | GNN-PPI | GNN-PPI+ | GNN-PPI | GNN-PPI+ | GNN-PPI | GNN-PPI+ |
| SHS27k | Random | 86.02 | 88.93 | 68.71 | 73.44 | 37.81 | 45.61 | 83.65 | 87.42 |
| | BFS | - | - | 64.83 | 73.18 | 41.44 | 56.79 | 63.08 | 69.52 |
| | DFS | - | - | 69.01 | 66.40 | 43.57 | 51.62 | 66.52 | 63.78 |
| SHS148k | Random | 91.97 | 91.57 | 70.75 | 79.50 | 43.90 | 59.60 | 90.87 | 87.33 |
| | BFS | - | - | 68.53 | 76.61 | 62.01 | 72.59 | 69.53 | 77.46 |
| | DFS | - | - | 69.33 | 73.52 | 49.31 | 57.40 | 75.34 | 79.58 |
| STRING | Random | 95.37 | 95.09 | 71.29 | 77.49 | 54.20 | 67.00 | 94.53 | 93.29 |
| | BFS | - | - | 67.91 | 75.66 | 42.08 | 63.02 | 75.69 | 79.91 |
| | DFS | - | - | 68.23 | 80.55 | 49.98 | 78.87 | 84.28 | 86.32 |

*Table 5.* More detailed results comparing GNN-PPI and GNN-PPI+ across the BS, ES, and NS subsets.

*Table 6.* Training efficiency on three benchmarks.

| Methods | SHS27k | SHS148k | STRING |
|---|---|---|---|
| PIPR | 679 | 3801 | 6500 |
| PIPR+ | 490 | 1693 | 2202 |
| GNN-PPI | 421 | 2258 | 3649 |
| GNN-PPI+ | 451 | 1982 | 2307 |
| MAPE-PPI | 127 | 1003 | 1935 |
| MAPE-PPI+ | 378 | 1864 | 1966 |

| Training methods | Time/s | Training Acc. | Testing F1 |
|---|---|---|---|
| Only P | **217** | 0.91 | 0.79 |
| Only G | 862 | 0.80 | 0.72 |
| P & G | 1424 | 0.93 | 0.83 |
| P → G | 451 | **0.98** | **0.87** |
| G → P | 989 | **0.98** | 0.84 |
| Iterative P&G | 1903 | 0.95 | 0.79 |

*Table 7.* Evaluation is performed on SHS27k with GNN-PPI as backbone. We define convergence as a fluctuation in the loss value of less than 0.05 over 10 consecutive epochs. The six training strategies, enumerated sequentially, comprise: (1) exclusive optimization of prompting parameters, (2) exclusive optimization of gating parameters, (3) simultaneous optimization both parameter sets, (4) sequential optimization with prompting parameters updated first followed by gating parameters, (5) sequential optimization with gating parameters updated first followed by prompting parameters, (6) alternating joint optimization of both parameter sets.

