# OpenReview forum: "Learning the Interaction Prior for Protein-Protein Interaction Prediction: A Model-Agnostic Approach"
_ICML.cc/2026/Conference — ICML 2026 regular_

### Official Review · Reviewer_2rZS · 2026-03-04

**Soundness:** 3
**Presentation:** 3
**Significance:** 3
**Originality:** 3
**Overall Recommendation:** 4
**Confidence:** 4

**Summary:**

To solve the challenge of incorporating biological complementarity priors in protein-protein interaction (PPI) prediction models, this works references the L3 rule and reformulates pairwise protein interaction prediction as a graph-level classification task over constructed pattern graphs. The proposed L3-PPI method employs graph prompt learning that aligns upstream and downstream tasks by pretraining a surrogate GNN model on L3 pattern graph validity recognition and then transforming PPI prediction into graph-level classification of prompted L3 structures, where K+1 virtual prompt nodes are introduced and shared between all query protein pairs with a gating mechanism determining the contribution of each candidate L3 path. L3-PPI functions as a lightweight, plug-and-play classification head that can be integrated into existing PPI predictors, replacing commonly used standard aggregators such as concatenation. Experiments on multiple benchmarks, covering both binary PPI prediction and interaction type prediction, demonstrate consistent performance improvements across diverse baselines.

**Compliance With Llm Reviewing Policy:**

Affirmed.

**Final Justification:**

The interpretability concern of this work is mostly addressed by the author rebuttal reply. Overall speaking, I am satisfied with the motivation, technical soundness and presentation of this work. I will keep my positive rating.

**Key Questions For Authors:**

1. The paper claims that L3-PPI offers better interpretability than existing GPL methods by following the L3 rule to construct prompts. I would like to clarify the scope of this interpretability. Does it refer to interpretability in design, meaning the rationale behind constructing prompts in this specific way?  Or does it extend to interpretability in the learned results, where activated L3 paths or virtual protein nodes correspond to identifiable biological entities? Could a case study be provided to illustrate this?
2. The main text refers to Appendix B, Table 6 for efficiency results, but the numbers in the table lack unit specification. And inference efficiency is not discussed.

**Limitations:**

Yes. The paper notes that the overall improvement depends on the baseline representation model, where gains are modest when the base model's representations are weak.

**Strengths And Weaknesses:**

Strengths
1. Focus on classification head in PPI prediction: Unlike many existing methods that primarily focus on learning powerful protein representations, this paper shifts the focus to the design of the classification head for PPI prediction and addresses the challenge of how to effectively use the extracted representations for final classification.
2. Solid biological motivation and robust injection of complementarity priors: The paper builds on a strong biological motivation by grounding the approach in the L3 rule and provides extensive empirical and statistical evidence supporting the L3 rule. Instead of directly using the number of L3 paths as a handcrafted feature, this method determines whether the constructed complementarity pattern graph resembles known valid L3 interaction structures, offering a more biologically plausible way to inject complementarity priors. Additional path number regularizing loss during training further reinforces this biological alignment. Moreover, it eliminates the dependence on graph structure disruptions caused by data partitioning during training and testing.
3. Clear methodological framework and presentation: The paper clearly outlines the graph prompt learning framework, detailing the upstream and downstream tasks. It effectively describes the process of using a graph as a prompt to transform a query protein pair into an L3-structured protein graph, along with the gating mechanism employed in this process.

Weaknesses
1. Limited improvement on some baselines: Gains on high-performing models like MAPE-PPI are marginal, raising the question of whether the added complexity is justified for already competitive approaches. Additionally, while the inclusion of path features and biological priors should theoretically improve models with weaker representation capabilities, the improvements in these cases are also modest.
2. Lack of theoretical guidance on K value selection: The choice of K determines the number of virtual nodes and the scale of the prompt graph. Although an empirical trend for selecting K is observed, the paper provides no theoretical justification for its value. It also lacks discussion on the biological meaning of K, for instance, whether the number of candidate L3 paths corresponds to any network property such as the average node degree in the actual PPI network. Without a clear rationale, the model’s generalization may be compromised when K is not appropriately tuned for a given baseline.

---

> ### Author Rebuttal · Authors · 2026-03-31
>
> We sincerely appreciate the reviewer 2rZS’s positive assessment of our work’s motivation, biological grounding, and methodological clarity. Below, we respectfully respond to the points raised.
>
> **[Cons. 1: Limited Gain for MAPE-PPI]** Thanks for raising this point. We apologize for an error in Table 1: we mistakenly used the pre-trained model instead of the supervised fine-tuned model of MAPE-PPI as the base for MAPE-PPI+. We have re-run the experiments using the fully trained MAPE-PPI model. As shown below, L3-PPI yields significant gains in the fully inductive BFS and DFS settings. In the Random setting, gains are relatively marginal but positive, likely because MAPE-PPI already performs exceptionally well and approaches the performance ceiling under the easy pattern (random setting).
>
>
> |SHS27k|Random |DFS|BFS|
> |-|-|-|-|
> |MAPE-PPI|88.91| 71.98| 70.38|
> |MAPE-PPI+|89.51|75.68|73.92|
>
>
> |SHS148k|Random |DFS|BFS|
> |-|-|-|-|
> |MAPE-PPI|92.38 |79.45| 74.76|
> |MAPE-PPI+|92.59|80.66|76.28|
>
>
> |STRING|Random |DFS|BFS|
> |-|-|-|-|
> |MAPE-PPI| 96.12 |86.50| 78.26|
> |MAPE-PPI+|96.70|87.43|79.65|
>
>
> Furthermore, we strongly agree with your intuition that L3-PPI provides more substantial boosts to models with weaker representations (e.g., DPPI, DNN-PPI, S2F in Table 1). The limited gain with SemiGNN-PPI is a notable exception. Because SemiGNN-PPI focuses purely on network topology and lacks structural/chemical features. Therefore, it may be difficult for L3-PPI to introduce or amplify its complementary knowledge. Additionally, its reliance on complex network dependencies (such as implicit L2 paths) may conflict with the L3 rule, hindering generalization. We will clarify this in the revised manuscript.
>
> **[Cons. 2: The $K$ Value]** We sincerely appreciate the insightful comment. Below, we provide some explanations.
>
> **Meaning of $K$:** $K$ represents the maximum expected complementary interface capacity between a protein pair, aligning with the L3 rule. From a learning view, $K$ also represents the upper limit of complementary interface information available for the model to learn regarding a specific protein pair $X$ and $Y$ within a given dataset. This is because, according to the L3 rule, a larger $K$ indicates that $X$ and $Y$ have a greater number of neighbors.
>
>
> **Selection of $K$:** As shown in Figure 6, the optimal $K$ scales positively with dataset size (i.e., the number of proteins). This is intuitively reasonable because: (1) Larger datasets naturally contain more L3 paths for positive PPI samples (averaging 149, 266, and 425 paths in SHS27k, SHS148k, and STRING, respectively). This indicates that the same protein pair possesses more complementary interface information available for learning in larger-scale datasets. If $K$ is set too small, it may create an **information bottleneck**. However, $K$ cannot be arbitrarily large, as an excessively high $K$ introduces **information redundancy**, which hampers smooth optimization and model convergence.
>
>
> **Conclusion:** If a dataset can provide more complementary interface information for each protein pair, we should consider appropriately increasing $K$ to accommodate this information gain, while an excessively large $K$ would lead to parameter redundancy.
>
> **[Q. 1: Interpretability]** Thanks for raising this valuable question. The interpretability of our GPL framework operates at the design level. In conventional GPL methods, the virtual nodes and topologies generated during prompting have no real-world significance because they only store implicit knowledge acquired during fine-tuning. While our prompting nodes also do not correspond to specific biological entities, the topological structures generated possess practical meaning in guiding model generalization. For example, for a given test protein pair, we can obtain the embedding of each virtual node in its prompted L3 paths, as well as the number of L3 paths, which can serve as an indicator of interaction likelihood. However, we have not trained an additional decoder to map these nodes back to concrete protein entities. To be honest, this is part of our ongoing work, which jointly explores PPI prediction and binder design to investigate how virtual nodes of the L3-PPI can be reconstructed into protein sequences.
>
> **[Q. 2: Efficiency]** Thanks for the kind reminder. This was an oversight on our part. The unit in Table 6 is seconds, and the values represent the time required for the model to be trained to convergence. It can be seen that although our method is integrated into existing PPI predictors, it shows no disadvantage in efficiency. This is because the trainable part of our model is lightweight: L3-PPI only takes the representations from the PPI predictor without optimizing the predictor itself.
>
> **We greatly appreciate the reviewer’s questions, as they all help improve the quality of the paper. All the results, discussions, and corrections mentioned above will be added to the manuscript.**

---

> > ### Author Rebuttal · Reviewer_2rZS · 2026-04-01
> >
> > I thank the authors for the detailed rebuttal and several of my concerns have been meaningfully addressed. In particular, the clarification on the corrected MAPE-PPI results strengthens the empirical claims, and the additional discussion on the role and interpretation of K provides useful intuition about its relationship to dataset scale and complementary interface capacity. The corrections regarding efficiency are also clear.
> >
> > However, interpretability concern remains only partially resolved. The authors clarify that interpretability primarily lies at the design level rather than in mapping learned structures to concrete biological entities. While this is reasonable, the current explanation still leaves a gap between structural interpretability and biological interpretability in practice. Particularly, it remains unclear to what extent the learned prompt structures (activated L3 paths or virtual nodes) correspond to meaningful or identifiable interaction patterns beyond serving as abstract indicators. Providing a more concrete case study or exploring mechanisms (such as a decoder) to relate these structures back to biological entities, would further strengthen this aspect and make the interpretability claim more compelling.

---

> > > ### Author Response · Authors · 2026-04-03
> > >
> > > We greatly appreciate the reviewer’s subsequent endorsement of our response and constructive suggestions. Over the past two days, we have designed additional experiments to address this issue.
> > >
> > > > **Why did our original manuscript only consider the design aspect?**
> > >
> > >
> > > We appreciate the reviewer’s acknowledgment of the reasonableness of the design-level feature in our method. Inspired by your comments, we would like to respectfully explain further why we only considered the design level before. The main reason is that, for Graph Prompt Learning (GPL), introducing interpretability in terms of statistical characteristics already constitutes a research gap. The development trajectory of GPL differs significantly from prompt learning techniques in Natural Language Processing (NLP), as the former lacks clear semantics akin to the latter.
> > >
> > > In NLP, prompts are based on understandable text, whereas advanced GPL methods typically augment the original graph/node with additional structure and node features whose information lacks direct interpretability. For example, in a citation network, prompt nodes are mere embedding vectors without corresponding real paper entities, and the prompt topology lacks domain meaning.
> > >
> > >
> > > In L3-PPI, while prompt nodes are also embedding vectors and not explicitly mapped to proteins, the prompt graph is fine-tuned to learn common L3-path patterns. This enables the model to associate high-probability PPI pairs with prompt graphs containing more L3 paths. This design offers a meaningful form of interpretability: during inference, users can inspect the number of L3 paths in the prompt graph as a consistency check for the confidence of PPI prediction.
> > >
> > > In summary, we respectfully elaborate on the non-trivial contributions of our GPL approach, while also **acknowledging the limitations** in interpretability regarding entity correspondence that you mentioned. We believe your suggestions have provided us with valuable insights.
> > >
> > >
> > > > **We verified that the original L3-PPI possesses a certain degree of biological entity association capability.**
> > >
> > > To evaluate whether the native L3-PPI captures entity-level information, we designed a test measuring the overlap between virtual prompt nodes/paths and real proteins/L3 paths in the PPI network. Specifically, we applied t-SNE to reduce the dimensionality and perform clustering on the embeddings of all prompt nodes for the query proteins as well as thei real neighboring proteins. We considered a match if a virtual node and a real protein fell into the same cluster. Furthermore, if the two nodes in a virtual L3 path and the two proteins in a real L3 path fall into the same cluster respectively, then the L3 path is considered to be reconstructed.
> > >
> > > We have the following results:
> > >
> > > |Coverage| SHS27k|SHS148k| STRING
> > > |-|-|-|-|
> > > |Node|0.462|0.466|0.513|
> > > |L3 Path|0.253|0.309|0.352|
> > >
> > > Note: Coverage (Node) refers to the proportion of virtual nodes in the prompt graph that coincide with actual neighboring proteins. Coverage (L3 Path) refers to the proportion of virtual L3 paths in the prompt graph that coincide with actual L3 paths.
> > >
> > > **Conclusion:** It can be seen that the native L3-PPI method generates virtual prompt nodes with satisfactory protein entity recovery capability. On average, for a given query protein pair, nearly half of the prompt nodes can be mapped to actual entities in the network. Importantly, it is not trivial, as the training process did not learn from any samples of the test set proteins, and the test set proteins themselves are isolated, with no neighbor information available for reference.
> > >
> > > > **How to further enhance the characterization of biological entities？**
> > >
> > > To further enhance the interpretability of our method at the entity level, we constructed an additional decoder. Specifically, we modified the inverse-folding model called ProteinMPNN to build the decoder, which takes the embeddings of the virtual prompt nodes as input to reconstruct the corresponding protein sequences. Moreover, to prevent training collapse, we considered selecting reconstruction targets from a larger-scale PPI network, STRING. Thus, we introduced an additional constraint during the training of L3-PPI, which encourages the generated virtual prompt nodes to reconstruct the neighboring proteins of the query protein in the STRING dataset. We found that this training branch slightly improves PPI prediction performance while significantly enhancing interpretability.
> > >
> > > |Coverage| SHS27k|SHS148k| STRING
> > > |-|-|-|-|
> > > |Node|0.575|0.562|0.713|
> > > |L3 Path|0.317|0.329|0.473|
> > >
> > > |Micro-F1| Random |DFS| BFS |
> > > |-|-|-|-|
> > > |GNN-PPI+| 87.42| 67.78| 69.52 |
> > > |GNN-PPI++|88.32|66.45|70.38|
> > > |PIPR+|83.22 |57.97 |49.02 |
> > > |PIPR++|83.53|58.69|51.37|
> > >
> > > Note: PIPR+ represents our native L3-PPI model. PIPR++ represents the one with the reconstruction regularization.

---

### Official Review · Reviewer_TPEe · 2026-03-07

**Soundness:** 2
**Presentation:** 3
**Significance:** 3
**Originality:** 3
**Overall Recommendation:** 3
**Confidence:** 3

**Summary:**

This paper proposes the L3-PPI framework, a plug-and-play graph prompt learning module. The paper provides empirical validation of the L3 rule and evaluates L3-PPI as an add-on to PPI predictors across multiple benchmarks.

**Compliance With Llm Reviewing Policy:**

Affirmed.

**Final Justification:**

After reading the authors’ response, I maintain my score.

**Key Questions For Authors:**

1.	The Pearson correlations between predicted and actual L3 paths (Figure 5B) are moderate. Could the authors discuss whether higher-fidelity path recovery would lead to further performance gains, or whether the current level of correlation is already sufficient for downstream prediction performance?

2.	The paper states that L3-PPI consistently improves over baseline methods. However, under Random partitioning the improvements appear modest and occasionally negative. Could the authors clarify under what conditions L3-PPI provides clear benefits and when it does not?

3.	The results suggest that L3-PPI performs poorly when the base representations are weak (e.g., SemiGNN-PPI). Could the authors provide a more systematic analysis of such failure cases?

4.	The reported results are averaged over three random seeds, but the main tables do not include standard deviations. Could the authors report variance statistics to better assess the stability of the results?

**Limitations:**

yes

**Strengths And Weaknesses:**

Strengths:

1.	The L3 rule appears biologically grounded and empirically validated.

2.	The plug-and-play design is a practical strength.

Weaknesses:

1.	The Pearson correlations between predicted and actual L3 paths (Figure 5B) are moderate, and it is worth discussing whether higher fidelity path recovery would yield further performance gains or whether this plateau is sufficient.

2.	The paper claims that L3-PPI consistently improves over baseline methods, but the improvements under Random partitioning are modest and occasionally negative. A clearer characterization of when L3-PPI does and does not help would strengthen the paper's practical guidance.

3.	L3-PPI struggles when base representations are weak (e.g., SemiGNN-PPI), the paper does not systematically analyze failure cases.

4.	Results are averaged over three random seeds, but no standard deviations are reported in the main tables.

---

> ### Author Rebuttal · Authors · 2026-03-30
>
> We sincerely appreciate the reviewer TPEe's recognition and suggestions. Below, we will address the reviewer's questions point by point.
>
> **[Cons. 1 Fidelity Path Recovery]:** We appreciate this very valuable comment. First, we present the answer to this question, followed by the detailed investigation process.
>
> **Conclusion:** We found that (1) the fidelity of path recovery can be controlled by adjusting the weight $\beta$ between the classification loss and the regularization term, i.e., $L_{total} = L_{BCE} + \beta L_{PN}$. (2) Higher fidelity of path recovery usually leads to better model performance.
>
> **Details:** First, we explore the relationship between fidelity of path recovery (FPR) and performance (micro-F1) on the test set. We sampled 20 model checkpoints near the best model and calculated their respective FPR and micro-F1 values on the test set. We found that the correlation coefficient between these two metrics is as high as 0.8579. Due to space limitations, we only show a subset of the data in the table below.
>
> |Checkpoints| 1|2| 3 |4|5|6|7|8|
> |-|-|-|-|-|-|-|-|-|
> |FPR|0.525|0.576|0.560|0.632|0.657|0.652|0.644|0.612|
> |Micro-F1(%)|84.78|85.31|85.09|86.33|87.30|87.42|86.72|86.05|
>
> As can be seen, the two metrics exhibit a very high correlation. Therefore, we consider how to further improve our method based on the reviewer's question, specifically by increasing the potential upper limit of FPR on the test set. Intuitively, the direct factor affecting FPR is the proposed regularization loss (Eq. 8 in the manuscript). Hence, we consider introducing a hyperparameter $\beta$ to balance $L_{BCE}$ and $L_{PN}$.
>
> We were surprised to find that this can bring improvement to L3-PPI. After hyperparameter search, experimental results are as follows:
>
> (Note: PIPR+ denotes the results reported in the manuscript, and PIPR++ denotes the results after introducing $\beta$.)
>
> |Method| Random |DFS| BFS |
> |-|-|-|-|
> |PIPR+| 83.22|  57.97|  49.02|
> |PIPR++|83.92 |58.12 |50.37 |
> |MAPE-PPI+| 87.93| 74.69 |74.50|
> |MAPE-PPI++| 87.86| 75.19|74.71 |
>
> In summary, this discovery further enhances our method and validates our paper's core contribution regarding the recovery of L3 paths.
>
> **[Cons. 2 Occasional Failure]:** Thanks for raising this point. We acknowledge that our method can be suboptimal in the random setting, as noted in Section 4.5 (line 328). However, we respectfully argue that the lower performance gain is relative to the DFS and BFS settings. Overall, our method still achieves a positive average gain of 1.05 under random splitting.
>
> We apologize for the lack of systematic analysis regarding this part. The reason our method shows a more significant advantage in the fully inductive setting (DFS/BFS) is that these protocols imply the test proteins are isolated, which leaves baseline methods unable to leverage any network topology. In this context, L3-PPI demonstrates a substantial advantage, as it has learned topological knowledge from the training data.
>
> We highly appreciate the suggestions for enhancing the quality of the paper and will incorporate the above discussion into the manuscript.
>
> **[Cons. 3 Base Representations are Weak]:** We sincerely thank the reviewer for this keen and insightful observation. The performance gains achieved when applying our method to the SemiGNN-PPI baseline are relatively limited compared to the significant boosts seen with other backbones.
>
> We provide a necessary explanation for the failure of our method on SemiGNN-PPI in the random setting. First, SemiGNN-PPI is a method that focuses primarily on network topology rather than protein representation, which may indeed result in weaker representations. This makes it difficult for L3-PPI to amplify complementary interaction information from them. Additionally, the complex network dependencies learned by SemiGNN-PPI may conflict with the L3 rule. For example, if it incorrectly and implicitly emphasizes L2 path information, it could have a detrimental effect on the generalization of L3-PPI.
>
> We highly value this point and will incorporate this analysis and discussion into the manuscript, which will enhance the systematic nature and readability of the paper.
>
> **[Cons. 4 Variance Statistics ]:** Thanks for this comment. We overlooked the need to present standard deviations and will include the complete table in the revised manuscript. Due to space constraints, we only show a portion here.
>
> |Method| Random |DFS| BFS |
> |-|-|-|-|
> |PIPR|0.62|4.21|4.05|
> |PIPR+|0.50|2.13|3.96|
> |GNN-PPI|0.40|1.58|4.99|
> |GNN-PPI+|0.45|0.92|2.90|
> |MAPE-PPI|1.52|1.60|3.77|
> |MAPE-PPI+|0.78|1.72|2.09|
>
> **Conclusion:** In the vast majority of cases, our method can effectively reduce the prediction uncertainty of the backbone, demonstrating the significance of the performance gain and the robustness of the proposed approach.

---

> > ### Author Rebuttal · Reviewer_TPEe · 2026-04-03
> >
> > Thank the authors for the responses. However, generalization to weak baselines is still a limitation. Therefore, I will retain my current score.

---

> > > ### Author Response · Authors · 2026-04-04
> > >
> > > Thank you for taking the time to review our response. We are pleased that the previous issue has been "fully resolved".
> > >
> > > Here, we acknowledge that the original version of L3-PPI underperforms in the random setup to a very small number of methods such as SemiGNN-PPI. We appreciate this observation and would like to clarify the context behind this result and provide additional experiments to address your concern.
> > >
> > >
> > > **[Explanation]:** Our proposed L3-PPI is a plug-and-play approach capable of providing an improved classification head for nearly all PPI predictors. While this demonstrates its practical utility, it also introduces evaluation challenges, as exhaustively validating L3-PPI on every baseline is infeasible. Therefore, we selected a limited yet representative set of 8 baseline methods and 4 pre-training methods. Even with this subset, considering the three datasets (SHS27k, SHS148k, and STRING) and three data splits (random, DFS, and BFS), a total of (8+4) × 3 × 3 = 108 full training and testing runs were required for L3-PPI. This scale naturally precludes independent hyperparameter tuning for each of the 108 experiments. Therefore, as noted in the appendix, we applied the optimal hyperparameters obtained from one specific model to all other settings, which led to the observed underperformance in 4 out of the 108 scenarios. Despite this limitation, L3-PPI still achieved a positive average improvement of +2.26 for SemiGNN-PPI across all evaluated settings.
> > >
> > >
> > >
> > > **[Conclusion]:** We would like to emphasize that our explanation is **NOT meant to avoid responsibility**. We fully acknowledge that this limitation existed in the original manuscript and represents a shortcoming of our initial evaluation. Our goal is to illustrate that in evaluating plug-and-play methods, occasional underperformance may stem from practical constraints such as experimental cost, rather than inherent model limitations.
> > >
> > > Furthermore, we would like to request the opportunity to present the results of L3-PPI with SemiGNN-PPI after comprehensive hyperparameter tuning. Over the past two days, we have conducted independent hyperparameter optimization for SemiGNN-PPI and have also included three additional traditional methods[1,2,3] that produce weak representations. The following experiments demonstrate that L3-PPI consistently delivers strong performance.
> > >
> > >
> > >
> > > |SHS27k| Random |DFS| BFS |Avg. Gain|
> > > |-|:------:|:---:|:---:|:---:|
> > > |SemiGNN-PPI|85.57|69.25|67.94|-|
> > > |SemiGNN-PPI+|87.65|77.52|73.07|+5.16|
> > > |[1]|79.02|40.11|38.79|-|
> > > |[1]+|82.64|42.73|42.06|+3.17|
> > > |[2]|71.10|43.87|48.05|-|
> > > |[2]+|74.62|44.83|52.74|+3.06|
> > > |[3]|73.06|42.48|54.61|-|
> > > |[3]+|75.98|48.70|57.29|+3.94|
> > >
> > > **References:**
> > >
> > > [1] Detection of protein-protein interactions from amino acid sequences using a rotation forest model with a novel pr-lpq descriptor. ICIC, 2015.
> > >
> > > [2] A method for predicting protein-protein
> > > interaction types. PLoS One, 2014.
> > >
> > > [3] Using support vector machine combined with auto covariance to predict protein–protein interactions from protein sequences. Nucleic Acids Research, 2008.

---

### Official Review · Reviewer_eWxo · 2026-03-13

**Soundness:** 4
**Presentation:** 4
**Significance:** 3
**Originality:** 3
**Overall Recommendation:** 4
**Confidence:** 4

**Summary:**

The paper propose L3-PPI, a model-agnostic framework that incorporates biologically grounded complementarity priors into protein-protein interaction (PPI) prediction through L3-path-regularized graph prompt learning. The method generate a prompt graph with virtual L3 paths based on protein representations and controls the number of paths. It reformulated PPI as a graph-level classification task and can be used as a lightweight, plug-and-play classification head across different PPI models.

**Compliance With Llm Reviewing Policy:**

Affirmed.

**Key Questions For Authors:**

See weaknesses.

**Limitations:**

L3 features may be not so useful for more advanced PPI prediction methods which already considered such topological features.

**Strengths And Weaknesses:**

Strengths:

1. It is well motivated and clearly written. The idea to transfer a plausible heuristics into an additional plug-and-play classification head is novel and practical.
2. Directly incorporating L3 rule (number of L3 paths) is challenging due to the train/test split, especially for the inductive learning setting. Using a gating network to select the active paths is technically sound.
3. It is compatible with different PPI backbones and shows consistent performance gain.

Weaknesses:

1. Although the L3 rule has some previous evidence, and showed empirical gain, I feel it is more like some simple features derived from the graph topology. I am wondering if adding other topological features directly into the prompt tuning can achieve similar effect. In addition, although the authors show the effect of each component (e.g. pretraining, gating) in the ablation study, there may be one more baseline that need to be compared with -- backbone LLMs with L3 path numbers as additional features.
2. The effect of adding L3 paths in the prompt framework maybe similar to using a enclosing subgraph for link prediction. So I would not rate its significance very high.  And it would be good to see the comparison with enclosing subgraph based method such as SEAL, or to see if it is still useful when using SEAL as backbones.

---

> ### Author Rebuttal · Authors · 2026-03-30
>
> We sincerely appreciate that Reviewer eWxo considered our idea well motivated, the manuscript clearly written, and the technology novel and practical. Below, we respectfully address Reviewer eWxo's questions point by point.
>
> **[Cons. 1: Adding other topological features]** Thanks for the great suggestion. We respectfully clarify that our prompt learning framework is specifically designed for the L3 feature and cannot be directly adapted to others (e.g., spectral, degree). Therefore, for a fair comparison, we use the GPF method [1] to generate enhanced node features, enabling baselines to handle the DFS/BFS scenarios.
>
> The baseline features are:
> * **Triadic Closure Principle (TCP)**: the number of L2 paths.
> * **Cycle Detection (CD)**: the number of cycles contained in the 3-hop subgraph of each central protein node.
> * **Degree**: the degree value of each protein node.
> * **Community**: the community index of each protein node
> * **Spectral**: the eigenvalue spectra of normalized Laplacian matrices of the whole PPI network.
>
> Experimental results: (Backbone: PIPR; Data splitting: DFS)
>
> |Method| SHS27k|SHS148k| STRING |Success|
> |-|-|-|-|-|
> |PIPR|52.19|61.38|64.97|-|
> |TCP|48.92|57.47|65.14|1/3|
> |CD|55.40|60.13|62.26|1/3|
> |Degree|52.05|58.93|67.23|1/3|
> |Community|49.85|59.87|61.19|0/3|
> |Spectral|55.38|62.19|62.44|2/3|
> |**L3**|**57.97**|**64.52**|**69.33**|3/3|
>
> **Conclusion**: The significant performance gap shows that effectively leveraging topological features requires a carefully designed prompt-tuning framework. Our core novelty lies not in simply using L3, but in designing a structured prompt learning framework that makes it work in fully inductive scenarios.
>
> **LLM+L3 Results (SHS27k):**
>
> We also integrated the L3 feature into a PLM-interact backbone (LLM-L3) and our full approach (LLM-L3-PPI).
>
> Experimental results on SHS27k:
> |Method| Random |DFS| BFS |
> |-|-|-|-|
> |PLM-interact|89.25|74.79|75.19|
> |LLM-L3|89.95|75.12|75.59|
> |**LLM-L3-PPI**|**90.16**|**76.71**|**76.70**|
>
> **Conclusion**: While LLMs have strong inherent capabilities, the L3 feature provides a clear boost. LLM-L3-PPI achieves the best results, especially in DFS/BFS scenarios, validating that our contribution extends beyond introducing the L3 rule.
>
> **[Cons. 2: SEAL]** We sincerely appreciate the reviewer's suggestions. We agree SEAL is an effctive method for solving **inductive** link prediction and its enclosing subgraph encompasses L3 information. However, in the challenging **fully inductive** PPI scenarios (DFS/BFS), test nodes have no observable neighbors, preventing SEAL from constructing its subgraph. To adapt it, we designed a graph prompt framework to generate a virtual enclosing subgraph (SEAL-Prompt). Thus, we compare with SEAL directly for random splitting, and with SEAL-Prompt for DFS/BFS.
>
> Experimental results on SHS27k:
> |Method| Random |DFS| BFS |
> |-|-|-|-|
> |PIPR|79.59 |52.19| 47.13|
> |PIPR-L3-PPI| 83.22| 57.97| 49.02|
> |SEAL|84.63|-|-|
> |SEAL-Prompt|84.44|66.28|64.29|
> |**SEAL-L3-PPI**|**87.31**|**71.13**|**69.55**|
>
> **Conclusion**: While SEAL was not originally designed for PPI prediction, it shows strong performance, surpassing PIPR. Our SEAL-Prompt adaptation works effectively in fully inductive settings. The superior results of SEAL-L3-PPI suggest that while the enclosing subgraph is related to L3 paths, it may include other features (e.g., L2 paths) that can be detrimental to PPI. Combining it with our L3-PPI framework yields the best results. We will add these results and discussions into our revised manuscript.
>
> **[Limitation: Other topological features]** Thank for the insightful comments. We entirely agree that other features could potentially enhance PPI prediction. Nevertheless, we respectfully highlight two irreplaceable advantages of our L3-PPI approach: **(1) Robust Validation:** L3 is a validated feature with significant positive gain for PPI. As shown in Figure 2 of the manuscript, protein pairs with >238 L3 paths have a 100% interaction probability. It is important to note that not all topological priors yield positive effects; for instance, the Triadic Closure Principlem, which is a common prior in social networks, has been shown to be detrimental to PPI tasks [2]. **(2) Fully Inductive Setting:** Most topological features fail to function under the fully inductive setting of PPI prediction. This setting is critically important for real-world applications, such as predicting interactions for newly discovered proteins that lack any known partners. A key contribution of our work is the proposed prompt-learning framework specifically designed for L3 features, which effectively addresses this limitation.
>
> We sincerely thank this comment and believe discussing this will encourage exploration of more effective features and corresponding frameworks.
>
>
> [1] Universal Prompt Tuning for Graph Neural Networks, NeurIPS 2023.
>
> [2] Network-based prediction of protein interactions, Nature Communications 2019.

---

> > ### Author Rebuttal · Reviewer_eWxo · 2026-04-03
> >
> > I appreciate the authors' effort to add new results (and even new modules), especially the SEAL variants. The paper is a boarderline. It proposed some interesting heuristic features but I still think it is not that significant since essentially its advantage can be covered by some path-based or subgraph-based link prediction methods. I will keep the score.

---

### Decision · Program_Chairs · 2026-04-30

**Decision:**

Accept (regular)

**Comment:**

This paper proposes a model-agnostic framework for learning interaction priors to improve protein–protein interaction (PPI) prediction, addressing an important and broadly relevant problem in computational biology. Reviewers find the core idea well-motivated and practically valuable, particularly the ability to enhance diverse backbone models without requiring architectural changes. The method is conceptually simple yet effective, and the empirical results demonstrate consistent improvements across multiple datasets and model families, supporting its generality. While some concerns were raised regarding the degree of novelty and the need for deeper analysis of learned priors and potential biases, these do not substantially detract from the overall contribution. The paper provides a useful and extensible approach that can be readily adopted by the community, and I support acceptance.